Predicting the geographic distribution of ancient Amazonian archaeological sites with machine learning

Walker Robert S. walkerro@missouri.edu 1
Ferguson Jeffrey R. 1 2
Olmeda Angelica 1
Hamilton Marcus J. 3
Elghammer Jim 1
Buchanan Briggs 4
1 Department of Anthropology, University of Missouri - Columbia , Columbia , MO , United States of America
2 Archaeometry Laboratory, University of Missouri Research Reactor Center, University of Missouri - Columbia , Columbia , MO , United States of America
3 Department of Anthropology, University of Texas at San Antonio , San Antonio , TX , United States of America
4 Department of Anthropology, University of Tulsa , Tulsa , OK , United States of America
Sponheimer Matt
Electronic publication date: 2023 Mar 31
Publication date: 2023
Volume: 11
Electronic Location ID: e15137
Received 2022 Dec 28; Accepted 2023 Mar 7
Copyright: ©2023 Walker et al.
Copyright year: 2023
Copyright holder: Walker et al.
License: This is an open access article distributed under the terms of the Creative Commons Attribution License, which permits unrestricted use, distribution, reproduction and adaptation in any medium and for any purpose provided that it is properly attributed. For attribution, the original author(s), title, publication source (PeerJ) and either DOI or URL of the article must be cited.
License URL: https://creativecommons.org/licenses/by/4.0/

Keywords: Amazonia, Archaeology, Geoglyphs, Deforestation, Agriculture, Remote sensing, Human-environment interactions, Random forest, Machine learning, Amazonian Dark Earths

Funding: The authors received no funding for this work.

==============================
Amazonia has as least two major centers of ancient human social complexity, but the full geographic extents of these centers remain uncertain. Across the southern rim of Amazonia, over 1,000 earthwork sites comprised of fortified settlements, mound villages, and ditched enclosures with geometric designs known as geoglyphs have been discovered. Qualitatively distinct and densely located along the lower stretches of major river systems and the Atlantic coast are Amazonian Dark Earth sites (ADEs) with deep anthropogenic soils enriched by long-term human habitation. Models predicting the geographic extents of earthworks and ADEs can assist in their discovery and preservation and help answer questions about the full degree of indigenous landscape modifications across Amazonia. We classify earthworks versus ADEs versus other non-earthwork/non-ADE archaeological sites with multi-class machine learning algorithms using soils, climate, and distances to rivers of different types and sizes as geospatial predictors. Model testing is done with spatial cross-validation, and the best model at the optimal spatial scale of 1 km has an Area Under the Curve of 0.91. Our predictive model has led to the discovery of 13 new geoglyphs, and it pinpoints specific areas with high probabilities of undiscovered archaeological sites that are currently hidden by rainforests. The limited, albeit impressive, predicted extents of earthworks and ADEs means that other non-ADE/non-earthwork sites are expected to predominate most of Western and Northern Amazonia.

Introduction

Amazonia harbors abundant archaeological evidence for ancient indigenous social complexity and organization, including monumental architecture (Prümers et al., 2022), garden cities (Heckenberger et al., 2003), road networks (Saunaluoma et al., 2021), earthworks/geoglyphs (Pärssinen et al., 2020), raised fields (Rodrigues, Lombardo & Veit, 2018), and anthropogenic soils (Woods & Denevan, 2009). Such impressive human niche construction and landscape modification, despite often poor-quality and acidic soils in difficult-to-clear rainforest environments, have been discovered even in uplands and interfluves once considered unfavorable to large populations (Meggers, 1971; Steward, 1946). Southwest Amazonia (Upper Madeira River) has over 1,000 earthwork sites clearly visible in satellite imagery (Pärssinen et al., 2020; Saunaluoma & Schaan, 2012). Hundreds more are likely still hidden under forest canopies (Saunaluoma, Pärssinen & Schaan, 2018). However, debates continue about whether ancient indigenous social complexity, niche construction, and habitat modifications were only locally extensive (McMichael et al., 2012; Piperno, McMichael & Bush, 2017) or existed more broadly across much of Amazonia, including the heavily forested interfluves in northern and western regions (WinklerPrins & Levis, 2020; Prümers et al., 2022).

Earthworks figure prominently into debates about the degree to which ancient humans modified environments. The most commonly occurring earthwork across the southern rim of Amazonia (Fig. 1) is the ditched enclosure in geometric designs known as the geoglyph. It took an estimated 8,000 human workdays to dig an average geoglyph (Pärssinen, Schaan & Ranzi, 2009). Such extensive amount of work may have had several proposed functions, including ceremonial, burial, residential, hydraulic, agricultural, and defensive (de Souza et al., 2018; Pärssinen et al., 2020). Excavations and surveys show that ditches are often situated inside embankments making defensive function less likely, and the generally limited amounts of ceramics and Amazonian Dark Earths (ADEs) inside enclosures support ceremonial or plaza-like functions rather than residential or agricultural (Saunaluoma & Schaan, 2012). Nonetheless, all six hypothetical functions are likely evidence of some degree of social complexity and sedentary populations. Beyond geoglyphs, the most visible indicators of sedentary populations, social organization, and large-scale cooperation (i.e., complexity) are sophisticated monument building, hydraulic earthworks, and fish weirs in zigzag lines in northern Bolivia (Erickson, 2000; Erickson, 2003; Prümers et al., 2022).

Figure 1 The three main types of Amazonian archaeological sites overlain with random spatial cross-validation blocks (k= 5 folds) used for model testing.

Earthworks dominate across the southern rim, whereas Amazonian Dark Earth sites (ADEs) are common along the lower stretches of the Amazon and other major rivers.

In addition to previously underestimated levels of social complexity, Southwest Amazonia is also the origin of lowland South America’s most important domesticated crop, manioc (Olsen & Schaal, 1999), as well as other plant domesticates such as peach palm, peanut, squash, gourd, rice, cocoyam, and chili pepper (Clement et al., 2015; Clement et al., 2016; Watling et al., 2015; Iriarte et al., 2020a; Lombardo et al., 2020). Southwest Amazonia may also be the homeland for the largest language family in terms of number of languages, Arawak (Walker & Ribeiro, 2011). It is the confirmed homeland for Tupí, the second largest language family in lowland South America (Walker et al., 2012). In sum, archaeological features like earthworks and domesticated crop and linguistic diversity all point to this region as an important Amazonian center of indigenous complexity, ingenuity, and invention.

In contrast to earthworks, Amazonian Dark Earth sites (Arroyo-Kalin, Neves & Woods, 2009; Clement et al., 2015) are scattered across the Amazonia and densely distributed on bluffs and uplands near large Amazonian rivers and on the eastern seaboard (Denevan, 1996; Kern et al., 2006). Recent surveys show a wider ADE distribution that includes interfluves (WinklerPrins, 2009; Franco-Moraes et al., 2019; WinklerPrins & Levis, 2020). ADEs are extremely fertile anthropogenic soils (Neves et al., 2003), and these rich soils are agriculturally productive to this day given their high levels of phosphorus, carbon, and nutrients. The deep soil horizons of ADEs are often replete with charcoal, ceramics, lithics, and other remains left by humans (Lehmann et al., 2006; Schmidt & Heckenberger, 2009). ADEs are indicative of social complexity given the long-term sedentism, extensive agriculture, and sheer expanse of the larger sites which can reach over 100 ha in size (Kern, De Lp Ruivo & Frazão, 2009). The occurrence of ADEs offer one line of evidence for ancient social complexity through larger and more sedentary populations, while earthworks offer another line of evidence through intensive work output and cooperation needed to dig ditches and build mounds in the creation of monumental architecture.

Some ADEs and earthworks began in the early Holocene (Iriarte et al., 2020b; Lombardo et al., 2020). However, the majority of both ADEs and earthworks date to 2,500-500 years BP (McMichael & Bush, 2019; Watling, Mayle & Schaan, 2018; Arroyo-Kalin & Riris, 2021). Across the southern rim of Amazonia, mounded ring villages appear to mostly replace geoglyphs starting around 1,000 years BP up until the devastating depopulation that occurred with European colonization (Iriarte et al., 2020b).

Previous distribution modeling exercises have separately examined only earthworks (McMichael, Palace & Golightly, 2014; de Souza et al., 2018) or only ADEs (McMichael et al., 2014; Palace et al., 2017) or summed both predicted probabilities together into a single model (McMichael et al., 2017; McMichael & Bush, 2019). These previous studies used maximum entropy models with random, pseudo-absence locations to contrast against occurrence locations. One important finding from previous work is that many geoglyphs (mostly in the state of Acre, Brazil) were built in or near semelparous bamboo forests, perhaps because forest clearing and burning are much easier after bamboo die-offs than in closed-canopy forests (McMichael, Palace & Golightly, 2014; Watling et al., 2017; Watling, Mayle & Schaan, 2018).

To better answer the broader question of the full geographic extent of various ancient Amazonian archaeological sites, we opt for multi-class classification models to allow earthworks, ADEs, and other non-earthwork/non-ADE sites to compete against each other in the prediction for each grid cell. Our models use spatial cross-validation folds to calculate performance metrics that help deal with spatial autocorrelation issues and distinguish amongst alternative models (e.g., changing spatial scales). This approach does not require random location sampling. The geographic distributions of earthworks, ADEs, and other archaeological sites are mostly distinct. As we experimented with different modeling approaches, we found drastic improvements in the generalizability of models with the inclusion of all three site types. Therefore, our approach is useful for making meaningful archaeological site predictions across all of Amazonia.

Material and Methods

We use locations for Amazonian archaeological sites from recent compilations (de Souza et al., 2018; WinklerPrins & Levis, 2021). The sample size totals are 510 ADEs, 1,100 earthworks, and 422 other archaeological sites (Fig. 1). Most earthwork sites are clearly visible in satellite imagery. Starting with an online dataset (http://jqjacobs.net/archaeology/geoglyph.html), we confirmed the geolocations of all 1,100 sites in satellite imagery, including the well-documented earthwork clusters in Acre (Saunaluoma, Pärssinen & Schaan, 2018), Upper Xingu (Heckenberger et al., 2003), Upper Tapajós (de Souza et al., 2018), and Llanos de Moxos in Northern Bolivia (Prümers et al., 2022). Ditched enclosures or geoglyphs are the most common type of earthwork site, followed by fortified settlements and mounded ring villages. Geoglyphs come in a variety of shapes and sizes, mostly circles, ovals, quadrangles, parallelograms, and D-shapes. Some geoglyphs co-occur with multiple adjacent shapes and inner and outer ditch systems (e.g., circle inside circle, circle inside square, etc.). Geoglyphs range in size from 31 m to 490 m in width, with a median width of 136 m, and original ditch depths of around 4–5 m (Saunaluoma & Schaan, 2012; Saunaluoma, Pärssinen & Schaan, 2018). Many geoglyphs are connected by complex networks of straight, walled roads, but we have considered them each to be separate sites for this analysis.

Machine learning

Machine learning is a proven approach to predictive analytics, leveraging large data sets with complex interrelationships (Bonetto & Latzko, 2020). Our application is a multi-class classification approach to specify and predict earthworks, ADEs, and other archaeological sites. We first evaluated several common machine learning algorithms including neural networks, extreme gradient boosting tree, support vector classifiers, and lasso logistic regression and found them to be relatively high performing. However, the random forest classifier consistently outperformed the other alternatives.

The random forest algorithm is known to give excellent classification results with fast processing speeds (Du et al., 2015). Random forests operate by constructing a multitude of decision trees (here the number is set at 1,000) and are therefore an ensemble supervised learning method. Some of the advantages of random forests are that they are robust to inclusion of irrelevant features, and they are invariant to transformations of predictor variables (Belgiu & Drăgu, 2016). For these reasons, the random forest classifier is popular for remote sensing data given its accuracy, speed, and ability to handle high data dimensionality and multicollinearity.

A total of 65 geospatial variables are used as predictors. A full list of variables with a brief description, source, spatial resolution, dataset, and code is available (https://doi.org/10.5281/zenodo.7651334). Climatic variables (n = 20), including temperature and precipitation and their various permutations, along with elevation, are from WorldClim version 2 (Fick & Hijmans, 2017) at 30 arc-second spatial resolution (approximately 1 km). Terrain ruggedness is at 1 km spatial grain (Amatulli et al., 2018). The regional soil phosphorus raster is specific to Amazonia at 0.25 arc-degree resolution (Quesada et al., 2010). Another 25 soil characteristics are from the Harmonized World Soil Database version 1.2 (Fischer et al., 2008) at 3 arc-minute resolution. The median values of geospatial predictors are used when the scale of analysis (see below) is coarser than the available resolution of the predictor variable, and simply extracted as a single value if the scale of analysis is at a finer scale than the available resolution. Distance to lakes, rivers, and coasts are calculated from the Global Self-consistent, Hierarchical, High-resolution Geography Database version 2.3.7 (Wessel & Smith, 1996). Black, white, and clear river systems are classified from the Amazon Aquatic Ecosystem Spatial Framework (Venticinque et al., 2016).

Machine learning algorithms are run in R (version 4.4.2; R Core Team, 2022) with the package caret (Kuhn, 2008). To evaluate the performance of the random forest algorithms, spatial cross-validation is run using five folds with each spatial block 500 km apart (Fig. 1). Spatial blocks are assigned to training and testing folds randomly and in an iterative fashion to better balance the three site type classes in each of the five folds using the blockCV package in R (Valavi et al., 2019). The performance metric is the area under the receiver operating curve (AUC) measured with spatial cross-validation. An uninformative model has an AUC of 0.5, or simply a guessing-chance classifier. Conventional random cross-validation unrealistically overestimates model performance (AUC > 0.99) because spatially structured data with random cross-validation has the effect of severely underestimating prediction error (Roberts et al., 2017). Spatial autocorrelation is strong with these archaeological locations. We tried removing predictor variables that were collinear (>0.8), but this has no effect on the spatial cross-validated AUC so we left in all variables. Likewise, reducing the overall number of predictor variables using recursive feature elimination in the caret package and the forward feature selection in the CAST package has limited effect on the AUC so we again decided to keep the full set of predictors. In both exercises there was no significant change to prediction maps (i.e., the models remained essentially the same). As mentioned above, the random forest classifier is an ensemble classifier that produces many decision trees using a randomly selected subset of training variables and so is generally robust to collinear and extraneous variables.

We use a 30 arc-second spatial sampling scheme to match the highest resolution of the bioclimatic and soil property rasters. In search of the optimal spatial scale for analysis, we then gradually increased the scale of our grids up to one whole arc-degree, taking the median value for each predictor variable across the grid cells and the modal site occurrence as the dependent variable. However, we found a consistent decline in spatial cross-validation AUC metrics at coarser resolutions. Therefore, we revert to the 30 arc-second scale which had the highest of all testing performance with an AUC of 0.91 (accuracy 0.88, mean sensitivity 0.83, mean specificity 0.90) using the spatial cross-validation scheme. This high resolution appears to be a better scale for analysis despite the increase in spatial autocorrelation because it generalizes better because of the larger sample size and higher resolution of the training data.

Our final model predicts site distributions for all of Amazonia using the same 65 geospatial measures of soil, climate, and distances to rivers, lakes, and shores. Probabilities for all three types of sites sum to one, and so for the counting of cells for each site type we assign the site type with the highest probability (i.e., higher than one third). Hotspots of high probability of undiscovered earthworks and ADEs are calculated using a modified form of predicted minus observed grid cells. Only those grid cells with a high predicted probability (>0.8) for either earthworks or ADEs are included but then removed if any observed site had previously been found on that grid cell. In addition, open (non-forest) habitats and deforested areas are also removed to focus on only those areas that are most likely to have undiscovered sites. Deforestation through the year 2021 is from Global Forest Watch version 1.9 (Hansen et al., 2013) at 30-m spatial resolution. A grid cell is considered deforested if over half its area had been cut down since the year 2000.

Results

Out of a total 65 geospatial variables, most of the top predictors are distances to major rivers of different sizes and types (especially black and clear rivers, Figs. 2 and 3). The top climatic variables are measures of temperature seasonality, considerably higher for earthworks given the increased seasonality of the southern rim of Amazonia. Seasonality is related to ease of forest clearing and burning given the approximately three-month dry season of the southern rim that contrasts with less seasonality in most of the Amazon Basin. Previous models also found temperature seasonality and related derivatives to be important predictors of earthworks (McMichael, Palace & Golightly, 2014; de Souza et al., 2018).

Figure 2 Several top predictors distinguishing the three types of archaeological sites include distance to largest rivers (primarily the Amazon), elevation, annual range in temperature, and levels of soil phosphorus.

Figure 3 Scaled variable importance measures for the random forest model.

Only the top 30 variables are shown (the bottom 35 are excluded here).

The best soil variable in our model is soil phosphorus, a known limiting factor for Amazonian biomass productivity favoring better agricultural potential, higher near the Andes and lower in central and eastern Amazonia (Reichert et al., 2022). Phosphorus is lowest for other archaeological sites perhaps pointing to its limiting factor in agricultural potential. Soil phosphorus is highest for earthworks showing the importance of better soils for these sites perhaps because they can support larger populations. Bamboo-dominated forests coincide with fertile soils (mostly eutrophic haplic cambisols and fluvisols) that cover much of Acre, Brazil (Quesada et al., 2010; Quesada et al., 2011; de Carvalho et al., 2013). ADEs are intermediate in phosphorus levels in Fig. 2, although phosphorus levels within the actual ADE soils themselves are much higher (Schmidt & Heckenberger, 2009; Asare, Afriyie & Hejcman, 2020).

In the spatially cross-validated confusion matrix, the model is slightly prone to confuse ADEs with other sites (4% misclassification rate averaged across spatial folds). The model is less likely to confuse earthworks with the other two site types (only 2% misclassified). This likely results from the tighter spatial clustering of earthworks making them more predictable in space. ADEs and other sites show some geographic overlap making it more difficult for the model to separate the two, although the overall model performance is still quite high.

The general geographic distributions of earthworks and ADEs show clear patterns with the model’s predictions (Fig. 4) and are consistent with previous models (McMichael, Palace & Golightly, 2014; de Souza et al., 2018; McMichael et al., 2014; Palace et al., 2017; McMichael et al., 2017). Earthworks are localized across the southern rim of Amazonia, whereas ADEs dominate along major rivers and the eastern seaboard. Less obvious from the original occurrence data is that other non-earthwork/non-ADE sites are predicted by the model to dominate essentially everywhere else, including most of western and northern Amazonia. In fact, summing up the number of cells using the model predictions show that other sites dominate more than two thirds of all Amazonia.

Figure 4 Random forest model predictions for each of the three site types.

Only specific locations exist where there are high probabilities of finding additional earthworks or ADEs that have yet to be discovered (Fig. 5). This search is narrowed down in three ways. First, we only consider areas with high predicted probabilities (>0.8) of finding earthworks or ADEs. Second, we exclude areas with currently observed sites. Third, we exclude currently non-forested areas using natural habitat and deforestation data as any sites here presumably are likely to have already been discovered. For both earthworks and ADEs, hotspots overlap with protected areas, not surprisingly since protected areas are mostly still forested. In fact, the numbered areas in Fig. 5 overlap with the following protected areas: (1) Parque Estadual Chandless, (2) Reserva Extrativista Ituxí, (3) Reserva Nacional de Vida Silvestre Amazónica Manuripi, (4) Parque Estadual Igarapés do Juruena, (5) Parque Estadual do Matupiri, (6) Parque Nacional do Jaú, (7) Floresta Nacional do Amazonas, and (8) Floresta Nacional de Seracá-Taquera.

Figure 5 Hotspots of high probability (>0.8) of undiscovered earthworks (1-4) and Amazonian Dark Earth (ADE) sites (5-8).

Plots are predicted minus observed results where raster cells with known sites, open (non-forest) habitats, and deforested areas removed. Recently discovered geoglyphs using our model’s predictions are marked with “X”s in areas 2 and 3. Inset image is one example of a large circular geoglyph (444 m diameter) recently discovered in satellite imagery.

With model probabilities as a guide to specific areas warranting further investigation, we found 13 geoglyphs that were not included in training data. These locations, and an example image of a geoglyph discovered in this way, are in Fig. 5. We were directed to the southern part of the state of Amazonas in Brazil off the Purus River (area 2) and to northern Bolivia between Madre de Dios and Abuna Rivers (area 3) by model predictions where pockets of recent deforestation are revealing geoglyphs. This type of exercise potentially leads to an iterative methodology: model predictions lead to discovery, and then more training data in turn improves the model.

Discussion

Machine learning classifiers add an important predictive tool to better geolocate the hidden remnants of ancient Amazonian archaeological sites. Our predictive model generates geographic hotspots for where we are likely to find earthworks and ADEs in the future. Given that the geographic distributions of ancient Amazonian archaeological sites are predictable, we use our model to pinpoint forested regions where earthworks remain to be discovered but are currently in forested areas and not (yet) visible with satellite imagery. We have had success discovering new geoglyphs using this methodology (Fig. 5). A corollary of the limited extent of earthworks and ADEs is that these may indicate changes in ancient human social complexity, and while certainly impressive and widespread, may be restricted to less than one-third of Amazonia, according to our model.

While it is possible that new finds will greatly increase the known extents of earthworks and ADEs across western and northern regions of Amazonia, our model finds this scenario unlikely. However, to make additional progress, ground truthing and newer technologies like LiDAR are also necessary. A combination of methods is needed to improve our ability to find and detect, and eventually to preserve and protect, the archaeological sites associated with such impressive social complexity in Amazonia. There is urgency to the preservation of these archaeological sites. We have noted numerous examples where intensifying development of county roads, state highways, cattle ranches, and agricultural fields have all degraded many geoglyphs over the last several decades.

Deforestation has revealed many ancient Amazonian archaeological sites, but obviously it would be preferable if future discoveries occurred without habitat destruction. LiDAR mapping of ancient archaeological features hidden under forest canopy has proven to be an important discovery tool. LiDAR successes in archaeology, such as for Mayan research, have come from high-resolution digital elevation models of archaeological sites that reveal micro-topography otherwise hidden by vegetation (Chase et al., 2011). While most known Amazonian geoglyphs have been exposed by recent deforestation, LiDAR is the clear choice for the future discovery of archaeological sites that are currently hidden within forested regions across Amazonia. Several geoglyphs were discovered in Acre, Brazil from LiDAR data originally purposed to develop aboveground biomass models in forests (Wagner et al., 2022), and we have now discovered 10 geoglyphs in this exact way (see examples in Fig. 6).

Figure 6 Example of a geoglyph complex discovered in LiDAR data (publicly available at https://daac.ornl.gov/CMS/guides/LiDAR_Forest_Inventory_Brazil.html) and processed using the lidR package in R.

There are two contiguous D-shaped geoglyphs connected to another barely visible geoglyph (left). The square (center) and rectangular (upper right) geoglyphs are connected by a straight, bermed road.

LiDAR studies are contributing to the growing evidence of Amazonian archaeological sites under vegetation or forest canopies (de Souza et al., 2018; Iriarte et al., 2020b; Stenborg, Schaan & Figueiredo, 2018; Prümers et al., 2022). Like our machine learning model, these LiDAR studies show expansion of settlements and structures extending considerably farther in geographic extent than what had been previously established through conventional fieldwork. Moreover, these LiDAR studies consistently show that settlement structures themselves are more complex and interconnected, through road networks for example, than what had been previously known. There is still a need to better understand the ancient cultural context of geoglyphs. While the limited amount of archaeological excavation data to date has proven informative, preserving largely undisturbed geoglyphs under forests might one day help provide additional information.

There is now a steady stream of research pointing to complex human–environment interactions in Amazonia, challenging notions of pristine forests and false human-nature or culture-nature dichotomies. Native Amazonians were transforming and domesticating forests and landscapes for thousands of years. Their societal development and ecosystem engineering techniques generated galactic regional polities, impressive monumental architecture, distributed urban networks, and lasting landscape legacies that are only now being more fully appreciated.

This article benefited immensely from advice, data, and help from James Jacobs, Gregorio de Souza, Oliver Coomes, Antoinette WinklerPrins, Carolina Levis, Juan Carlos Fernandez Diaz, Marcus Vinicio Neves de Oliveira, Fabien Wagner, Crystal McMichael, Jennifer Watling, and the AmazonArch group.

Additional Information and Declarations

Competing Interests

Author Contributions

Data Availability

The authors declare there are no competing interests.

Robert S. Walker conceived and designed the experiments, performed the experiments, analyzed the data, prepared figures and/or tables, authored or reviewed drafts of the article, and approved the final draft.

Jeffrey R. Ferguson conceived and designed the experiments, authored or reviewed drafts of the article, and approved the final draft.

Angelica Olmeda performed the experiments, analyzed the data, authored or reviewed drafts of the article, and approved the final draft.

Marcus J. Hamilton conceived and designed the experiments, analyzed the data, authored or reviewed drafts of the article, and approved the final draft.

Jim Elghammer conceived and designed the experiments, authored or reviewed drafts of the article, and approved the final draft.

Briggs Buchanan conceived and designed the experiments, authored or reviewed drafts of the article, and approved the final draft.

The following information was supplied regarding data availability:

The data and models are available at Zenodo: Robert S Walker. (2023). RobertSWalker/ancient_amazonia_archaeology: v1.0.0 (v1.0.0). Zenodo. https://doi.org/10.5281/zenodo.7651334.

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
