# Peer review of "Predicting the geographic distribution of ancient Amazonian archaeological sites with machine learning"

_PeerJ, doi:10.7717/peerj.15137_

## Round 0.1 · original submission · Major Revisions

This is a nice contribution. The reviewer comments, in balance, seem to be quite justified. And while there is quite a bit to address, I think most of the comments can be dealt with quite easily. I really see this as somewhere between minor and major revisions.

Please address reviewer concerns in your manuscript and/or revision letter. I look forward to seeing the revised version of this very interesting manuscript.

Reviewer 1 ·

Basic reporting

A well written paper, focused on the use of machine learning methods (random forest classifer) to predict new archaeological sites of ancient Amazonian. Data, methodological steps and assumptions are provided in the text.
References are provided will the conclusions are also supported by the material.

Experimental design

Please elaborate more on the issues of the different variables used for the prediction model, especially the issue of the different scales of these parameters. As the variables have different resolutions (e.g. pixel sizes), how did you overcome this issue? In addition, please provide some details regarding the training and validation sample of the random classifier.

Validity of the findings

Very interesting findings. Please provide more details on the results of the random forest, such as a confusion matrix from the random forest classifier.

Additional comments

An interesting topic with excellent structure and material used. Some comments are provided to the authors only in terms of improvements.

·

Basic reporting

The paper is overall clearly written, but there are a number of lapses in the English language – particularly in sentence structure – that sometimes make the meaning of the text ambiguous. I list these below:
p.87: “more complex and sedentary populations” – more complex and sedentary than what?
p.99: “the largest language family” – largest in terms of what? Number of languages pertaining to it, number of people who speak it?
p.102: “this region as the lowland´s center of indigenous complexity, ingenuity and invention” – Noone would doubt that that these things all took place in SW Amazonia, but to say it was THE center of all complexity in all of lowland Amazonia (and/or South America, since it is unclear what is meant by lowlands) makes little sense and is frankly impossible to argue.
p.106: Again, the use of “lowlands” here – do you mean lowland Amazonia or South American lowlands?
p.110: ADEs are not renowned for their extremely fertile anthropogenic soils, they ARE extremely fertile anthropogenic soils.
p.119: This is the first time I have come across the term “Late Cultivation Period” – please provide a specific reference for this.
The references cited, as well as the number and quality of figures used, are appropriate.

Experimental design

As the authors mention, previous studies have modelled the distributions of earthworks and ADEs in Amazonia, but never both together. The relevance of this approach becomes more obvious in the study´s findings, since it has made possible the separation of different environmental “preferences” for different site types (e.g. Figure 2), which is an interesting result.
I have only two concerns (the first is actually a doubt) regarding this aspect of the paper:
1) If I am not mistaken, such “environmental preferences” revealed by the study are not implicitly understood as explanatory factors for the presence or absence of these sites in a given area, or are they? As someone with limited knowledge of the method, I wonder: how much of the model´s outcomes could be explained by the fact that, in SW Amazonia, there existed a series of cultures who, for one reason or another (and unlike their central and eastern Amazonian contemporaries), liked building ditched earthworks? How are these “cultural differences”, which are also reflected in the different archaeological ceramic traditions, perceived by the authors, and indeed by the model?
2) When the research question or relevance of the paper is “sold” on the basis of understanding Amazonian social complexity, I feel very strongly that this should not be the case (see the “Additional comments” section).

The methods are explained in detail and ethical standards appear to have been maintained. The only doubt I have is regarding the source of the Lidar photo in Figure 5, since it is not clear from where/whom this was obtained, and what permissions were sought to capture and publish this image. Please clarify.

Validity of the findings

The findings appear to be valid and meaningful. I am unable to judge the robustness of the method or results since I am not familiar with the methodology.
In terms of the conclusions, I have serious concerns regarding the comments made about Amazonian "social complexity" which I feel must be addressed (see the "Additional comments" section below).

Additional comments

I am one of a substantial number of archaeologists that no longer uses the term “complex” societies due to its complicated evolutionary overtones and the fact that it is poorly defined and generally not very useful for thinking about how past societies actually organized themselves. In most of the text I have ignored its usage: however, there are places where it is part of the paper´s arguments or conclusions which, in my opinion, makes it much more problematic. In each of these cases, I suggest the authors either get rid of this type of language altogether or, if they want to really make the argument, be far more detailed and explicit (with references) about what is meant by “more” or “less” complex societies. For example:
- p.121: The authors cite the replacement of geoglyph sites by ring villages as an indicator of “increasing social complexity”. There is no evidence to support this claim. If anything, the monumentality of the geoglyphs is of a far greater scale than that of the mound villages.
- p.135: Here the authors state that better knowledge of the spatial distribution of archaeological sites will allow one to “better answer the broader question of the full extent of ancient indigenous social complexity”. Are more densely-packed sites synonymous with higher social “complexity”? This is very hard to argue.
p.148-150: Here we are suddenly introduced to the concept of complex “sites” without any indication as to what this may mean
p.304: “ancient human social complexity.. is restricted to less than one-third of Amazonia according to our model”… There is absolutely no theoretical or methodological baseline for classifying non-ADE/earthwork sites as remnants of less-complex indigenous societies, and I feel strongly that such a comment should not be published. To back up this claim, the authors do not even attempt to offer details surrounding the chronologies of these sites, or the material culture they contain (whether they are different ceramic traditions to those found in ADE or earthwork sites, for example). We cannot discard the hypothesis, for instance, that many of these non-ADE/earthwork sites were occupied by the very same populations who occupied ADE and earthwork sites, since 1) they have very close, often overlapping, distributions (see Figure 1), 2) many geoglyphs, for instance, weren´t inhabited, meaning that people actually lived in other places, and 3) such sites could easily represent temporary, seasonal occupations (see ethnographic literature on amerindian mobility patterns).
p.344-346: Please see comment above about social complexity and change accordingly. Furthermore, it is inappropriate and scientifically inaccurate to say that niche constructing activities only occurred in the past where there are earthworks and ADEs.... Please be careful when making such sweeping comments, as they can have huge (if unintended) negative implications for the rights of indigenous groups living in the Amazon today.

·

Basic reporting

This manuscript is for the most part clearly written and has the potential to be a valuable contribution to understanding how land use varied across the Amazon during the pre-Columbian period. The authors use a machine learning model (random forest) to predict the distributions of three types of archaeological sites (ADE, earthworks, and non-ADE/non-earthwork) across the Amazon. However, the authors are lacking several key references that have performed similar methodologies and obtained similar results. McMichael et al. (2017, reference pasted below) summed the probabilities of the ADE, earthworks, and other types of archaeological sites. These estimates were derived from maximum entropy (Maxent) models, which, like the random forest models, are a machine learning approach. McMichael & Bush (2019) and Sales et al (2022) (full references pasted below) have used ensemble modeling techniques that include the random forest model (alongside other modeling algorithms) to make predictions on the distributions of people across Amazonian and Andean landscapes.

The comparison of the authors’ random forest model with these previously published model outputs would significantly strengthen the manuscript.

Experimental design

It is nice to see the addition of new predictor variables used in this study compared with the previously published work, especially those that are describing the size and type of river. It is also nice to see the spatial cross-validation used to test the model in this study. However, Meyer et al (2019) and others have shown that autocorrelated predictors variables can affect the output of Random Forest models. It is unclear whether the authors explored the effect of autocorrelated predictor variables, but it would be nice to see comparisons of the outputs of models using the full array of predictors and models using a reduced number of variables based on autocorrelation values as a supplement to this manuscript.

The authors mention the spatially autocorrelated distribution of their response variable (occurrence of sites), and the text implies that the spatial cross-validation accounts for the autocorrelated structure of the occurrence points (Lines 207-208). The spatial cross-validation of the model is likely to be affected by the high densities of occurrence points in some regions and low numbers of occurrence points in other regions. This is true especially in the non-ADE/non-earthwork category (Fig. 1), and sampling bias is likely to play a role in the documented distribution (McMichael et al. 2017). This geographic sampling bias of occurrence records can also affect model results, and filtering of the occurrence dataset is recommended (see Feng et al 2019 and references within). It is possible, or even likely, that the lack of spatial filtering on the occurrence datasets results in the very high overall values seen in the non-ADE/non-earthwork category. It would be nice to see the results of spatially filtered models in comparison with those used here as a supplementary figure/analysis. The authors also mention on Lines 138-140 that “In previous studies…. Also, our approach does not require random location sampling.” The previous studies referred to some of McMichael’s previous work that used maximum entropy as opposed to random forest algorithms. However, both algorithms are based on machine learning and are robust to non-random sampling, especially when accounting for spatial autocorrelation in the occurrence points and predictor variables.

The authors state, “Previous modeling exercises have separately examined only earthworks (McMichael, Palace & Golightly, 2014; de Souza et al., 2018) or only ADEs (McMichael et al., 2014; Palace et al., 2017) but not together in the same model.” (Lines 126-128). McMichael et al. (2017) combined the probabilities for similarly constructed predictive models of earthworks (McMichael et al. 2014a), ADE (McMichael et al. 2014b) and other types of sites and includes some, but not all, of the sites used in this paper. As a result, the outputs shown in Fig. 4a are very similar to those reported for earthworks in McMichael et al. (2014a). The predictions in Fig. 4b for ADE are also similar to previously published work (McMichael et al. 2014b). In this study, Fig. 4c shows the likely distribution of non-ADE/non-earthwork sites, and is similar to the cumulative model in all site types from McMichael et al. 2017. This paper would be significantly strengthened if comparisons with these previously published models, which have spatially filtered occurrence data and used an alternative machine learning algorithm (maximum entropy as opposed to random forest).

Validity of the findings

The results section (Lines 239-257) report summary trends and characteristics for all three models, but do not directly compare how the values for the different predictor variables vary between ADE, earthwork, and non-ADE/non-earthwork sites. Reporting these comparisons between the three models would be a valuable contribution to the literature (and was a main focus of the paper).

Lines 268-272 of the results suggest that non-ADE/non-earthwork sites ‘dominate’ certain areas of Amazonia. This statement also appears as the last sentence of the abstract (Lines 52-54). This cannot be concluded unless the ‘overprediction’ aspect of having autocorrelated predictor variables and occurrence points is addressed (see Experimental Design section above).

The authors state that “The geographic distributions of earthworks, ADEs, and other archaeological sites are fairly distinct. We found drastic improvements in the generalizability of our model with the inclusion of all three site types. Therefore, we are able to make meaningful archaeological site predictions across all of Amazonia.” (Lines 140-143). To find drastic improvements in the generalizability of their model for these three site types, the authors would have needed to make direct comparisons with the ADE model of McMichael et al. 2014a, the earthwork model of McMichael et al. 2014b, and the overall model of McMichael et al. 2017 (also see Basic Reporting and Experimental Design sections).

Additional comments

The discussion would also benefit from some text on the comparisons with the previously published models of earthworks, ADE, and other types of archaeological sites. Please see the following references regarding statements made in the first sections. References with asterisks(*) are not included in the manuscript, and references without asterisks are included in the manuscript and in the above review.

*Feng, Xiao, et al. "A checklist for maximizing reproducibility of ecological niche models." Nature Ecology & Evolution 3.10 (2019): 1382-1395.

McMichael, Crystal H., et al. "Predicting pre-Columbian anthropogenic soils in Amazonia." Proceedings of the Royal Society B: Biological Sciences 281.1777 (2014): 20132475.

McMichael, Crystal H., Michael W. Palace, and Megan Golightly. "Bamboo‐dominated forests and pre‐Columbian earthwork formations in south‐western Amazonia." Journal of Biogeography 41.9 (2014): 1733-1745.

*McMichael, Crystal NH, et al. "Ancient human disturbances may be skewing our understanding of Amazonian forests." Proceedings of the National Academy of Sciences 114.3 (2017): 522-527.

*McMichael, Crystal NH, and Mark B. Bush. "Spatiotemporal patterns of pre-Columbian people in Amazonia." Quaternary Research 92.1 (2019): 53-69.

*Meyer, Hanna, et al. "Importance of spatial predictor variable selection in machine learning applications–Moving from data reproduction to spatial prediction." Ecological Modelling 411 (2019): 108815.

*Sales, Rachel K., et al. "Potential distributions of pre-Columbian people in Tropical Andean landscapes." Philosophical Transactions of the Royal Society B 377.1849 (2022): 20200502.

---

## Round 0.2 · accepted · Accept

Good faith effort here. I'm inclined to accept as is rather than go back to the reviewers.